# RA-SpaRC: Robust Adaptation with Sparse plus Low-Rank Compressors

## Abstract

Parameter-Efficient Fine-Tuning (PEFT) methods, such as Low-Rank Adaptation (LoRA), are widely adopted for their efficiency. However, LoRA assumes model updates are inherently low-rank, which introduces a restrictive bias that results in underperformance compared to full fine-tuning. Hybrid approaches, such as Robust Adaptation (RoSA), improve expressiveness by combining low-rank and sparse components, but they rely on a manually tuned ratio to balance these components, leading to suboptimal parameter allocation across tasks. We introduce RA-SpaRC (Robust Adaptation with Sparse plus Low-Rank Compressors), a new initialization strategy that overcomes this limitation. The key idea is an adaptive allocation mechanism that automatically balances sparse and low-rank components within a given parameter budget. This approach removes the need for manual rank–sparsity tuning and supports arbitrary parameter budgets. This principled and automated design allows RA-SpaRC to consistently outperform LoRA, its variants, and RoSA in extensive experiments across multiple models, delivering more effective and flexible adaptation.

## 1 Introduction

The rapid advancement of large language models (LLMs) and foundation models has revolutionized various domains in artificial intelligence, enabling remarkable performance across tasks such as natural language understanding and generation (Touvron et al., 2023; Radford et al., 2021). However, the sheer scale of these models, often comprising billions of parameters, poses significant challenges for fine-tuning on downstream tasks. Full fine-tuning (FFT) of all parameters is computationally intensive and memory-prohibitive. This has driven the need for parameter-efficient fine-tuning (PEFT) methods, which optimize a small subset of parameters while keeping the original pretrained weights frozen (Houlsby et al., 2019; Hu et al., 2022).

Among parameter-efficient fine-tuning (PEFT) techniques, Low-Rank Adaptation (LoRA) and Sparse Adaptation have gained prominence due to their simplicity and effectiveness (Hu et al., 2022; Sung et al., 2021). LoRA approximates weight updates as the product of two low-rank matrices, while sparse adaptation updates only a small subset of parameters. Both methods significantly reduce the number of trainable parameters, but can exhibit a performance gap relative to full fine-tuning (Wang et al., 2024; Sung et al., 2021). This gap stems from their reliance on low-rank or sparse approximations, which may not fully capture the intrinsic structure of weight updates in pretrained models.

To bridge this limitation, hybrid PEFT approaches that combine low-rank and sparse adaptations have emerged. For instance, Robust Adaptation (RoSA) (Nikdan et al., 2024) jointly trains low-rank and sparse adapters on top of fixed pretrained weights, drawing inspiration from Robust Principal Component Analysis (RPCA) (Candès et al., 2011) to decompose updates into low-rank and sparse components. The key advantage of this approach lies in the complementary nature of the components: sparse matrices are typically high-rank, whereas low-rank matrices are typically dense; thus, integrating them leverages their respective strengths.

The initialization strategy of PEFT methods is critical for both low-rank (Wang et al., 2024; Zhang et al., 2025; Meng et al., 2024) and sparse adaptations (Sung et al., 2021; Fu et al., 2023). Given a fixed parameter budget, methods like RoSA (Nikdan et al., 2024) and DSEE (Chen et al., 2021b) have to pre-define the rank and sparsity for each layer. This fixed allocation is often inefficient and

creates a difficult trade-off. Over-allocating resources to the low-rank components might neglect important sparse outliers, whereas an excessive sparsity level can compromise the power of adapters to capture global updates. This dielmma raises a natural and important question:

*"Is there an initialization strategy that enables flexible, automatic, and effective budget allocation for Robust Adaptation?"*

In this paper, we introduce RA-SpaRC (**R**obust **A**daptation with **Spa**rse plus Low-**R**ank **C**ompressors), an initialization strategy for sparse plus low-rank fine-tuning, which could dynamically assign the ratio of low-rank and sparse parts according to the gradient information of different tasks and models. Our method provides an efficient, data-driven solution to the static allocation problem, improving performance without increasing the parameter budget.

**Our contributions are summarized as follows:**

- We propose a unified framework for initializing PEFT methods based on compressors, and we show that under proper settings, this initialization indeed guarantees a loss decrease.
- Within this framework, we introduce sparse plus low-rank compressors for Robust Adaptation. We formulate the task of getting the compressed results as an optimization problem and propose an efficient algorithm to solve it.
- We demonstrate the efficacy of RA-SpaRC on both natural language understanding and natural language generation tasks.

## 2 RELATED WORK

**Parameter Efficient Fine-Tuning:** Parameter-Efficient Fine-Tuning (PEFT) methods adapt large models by training only a small fraction of their total parameters. A prominent example is Low-Rank Adaptation (LoRA), which approximates the weight updates using low-rank matrices (Hu et al., 2022) The performance of LoRA is known to be sensitive to its initialization, leading to recent works on more sophisticated initialization schemes such as LoRA-GA (Wang et al., 2024), LoRA-One (Zhang et al., 2025) and PiSSA (Meng et al., 2024). To overcome the expressive limits of the low-rank hypothesis, hybrid methods like DSEE (Chen et al., 2021b) and RoSA (Nikdan et al., 2024) combine low-rank updates with sparse updates. In contrast to their reliance on a fixed, pre-defined allocation of the parameter budget, RA-SpaRC determines this allocation dynamically.

**Robust Principal Component Analysis:** Robust Principal Component Analysis extends classical PCA to handle data corrupted by outliers or gross errors, decomposing a matrix into a low-rank component plus a sparse outlier matrix (Wright et al., 2009). The problem can be represented as:

$$\min_{L,S} \quad \text{rank}(L) + \tau \|S\|_0$$
$$\text{s.t.} \quad \|S + L - M\|_F \leq \delta. \tag{1}$$

Directly solving this optimization problem is NP-hard due to the non-convex nature of the rank function and the $\ell_0$-norm. Therefore, a common approach is to consider its convex relaxation (Chandrasekaran et al., 2011; Candès et al., 2011), where the rank function is replaced by the nuclear norm ($\|L\|_*$) and the $\ell_0$-norm by the $\ell_1$-norm ($\|S\|_1$). This relaxed convex problem can then be solved efficiently using Alternating Direction Methods (Tao & Yuan, 2011; Yuan & Yang, 2013).

**Compressors:** Our initialization framework is built upon operators from the field of compression. Compressors are developed to reduce communication overhead in distributed training (Li et al., 2022; Chen et al., 2023), and improve the memory efficiency of optimizers (Modoranu et al., 2024). There are two prevalent classes of compressors: sparse compressors like TopK, which preserve the $k$ largest-magnitude elements (Aji & Heafield, 2017; Lin et al., 2017), and low-rank compressors like SVD, which project the gradient onto a low-rank subspace (Vogels et al., 2019; Wang et al., 2018). These have been studied extensively, but almost always in isolation. Our primary technical contribution is the formulation of new hybrid compressors designed specifically for model initialization.

## 3 METHOD

This section details our proposed method. We begin by establishing a formal framework that unifies recent PEFT initialization techniques under the concept of compressors. We then introduce SpaRC (**Spa**rse Plus Low-**R**ank **C**ompressors), a novel hybrid operator designed to find an optimal sparse plus low-rank decomposition of the gradient for a given parameter budget. We present an efficient algorithm for this decomposition and demonstrate how it is used to initialize the PEFT adapters in a single step.

### 3.1 UNIFYING PEFT INITIALIZATION VIA COMPRESSORS

The initialization of many Parameter-Efficient Fine-Tuning (PEFT) methods can be conceptualized as applying a compressor to the full gradient. We define a compressor as an operator that approximates a high-dimensional gradient matrix with a low-parameter structure. Formally, we define this in the space of matrices equipped with the Frobenius norm.

**Definition 1** (Compressor). *The mapping $\mathcal{C} : \mathbb{R}^{m \times n} \to \mathbb{R}^{m \times n}$ is called a compressor if there exists a constant $\alpha \in (0, 1]$ such that for any matrix $X \in \mathbb{R}^{m \times n}$:*

$$\|\mathcal{C}(X) - X\|_F^2 \leq (1 - \alpha)\|X\|_F^2, \tag{2}$$

*where $\|\cdot\|_F$ denotes the Frobenius norm.*

In our context, the matrix $X$ represents the unbiased stochastic gradient of the loss with respect to the weight matrix $W_{0,\ell}$ of $\ell_{th}$ layer, which we denote as $g_{\xi,\ell} \overset{\text{def}}{=} \nabla_{W_{0,\ell}} \mathcal{L}(W_0; \xi)$, where $W_0 \overset{\text{def}}{=} (\cdots, W_{0,\ell}, \cdots)$ represents the parameters of all such layers, $\mathcal{L}$ is the loss function and $\xi$ is a mini-batch of data. We also denote $g_\xi \overset{\text{def}}{=} (\cdots, g_{\xi,\ell}, \cdots)$ as the whole unbiased stochastic gradient, $g \overset{\text{def}}{=} \mathbb{E}[g_\xi] \overset{\text{def}}{=} (\cdots, \mathbb{E}[g_{\xi,\ell}], \cdots)]$ as the true gradient, and $g'_\xi \overset{\text{def}}{=} \mathcal{C}(g_\xi) \overset{\text{def}}{=} (\cdots, \mathcal{C}(g_{\xi,\ell}), \cdots)$ as the whole compressed stochastic gradient. Besides, we define a new type of inner product $\langle X, Y \rangle = \sum_\ell \text{Tr}(X_\ell^T Y_\ell)$ and norm $\|X\|^2 = \langle X, X \rangle$ for both $W_0$, $g$, $g_\xi$ and $g'_\xi$. With this formal definition, we can now categorize the initialization strategies of popular PEFT methods:

- **Low-Rank Compression (SVD):** Methods like LoRA-One (Zhang et al., 2025) initialize the update by computing the best rank-$r$ approximation of $g_{\xi,\ell}$. By the Eckart-Young-Mirsky theorem, this is achieved via Singular Value Decomposition (SVD). This SVD$r(\cdot)$ operator is a projection onto a lower-dimensional subspace and is a well-known compressor that satisfies Definition 1 with $\alpha = \frac{r}{\min\{m,n\}}$.

- **Sparse Compressors (TopK):** Sparse methods like FISH-Mask (Sung et al., 2021) initialize the update by retaining only the $k$ largest-magnitude elements of $g_{\xi,\ell}$. This Top$_k(\cdot)$ operator produces a sparse matrix where all other elements are zero. It is also a powerful compressor that adheres to Definition 1 with $\alpha = \frac{k}{mn}$, as it preserves the most significant components of the gradient signal.

The insight that PEFT initializations can be viewed as compressors allows us to generalize the update rule. The change in weights, $\Delta W$, can be expressed as the application of a compressor $\mathcal{C}$ to $g_\xi$:

$$\Delta W = -\eta \cdot \mathcal{C}(g_\xi), \tag{3}$$

where $\eta$ is a learning rate or scaling factor. This single equation elegantly encompasses both low-rank adaptation where $\mathcal{C} = \text{SVD}_r(\cdot)$ and sparse adaptation where $\mathcal{C} = \text{Top}_k(\cdot)$.

### 3.2 SPARC: A HYBRID SPARSE PLUS LOW-RANK COMPRESSOR

Framework 3 provides a clear path forward for more complex PEFT structures. For methods like Robust Adaptation, which require a parameter-efficient update that is simultaneously sparse and low-rank ($\Delta W = \text{Low-Rank} + \text{Sparse}$), we must design a compressor that can extract both types of information from the gradient.

To this end, we propose SpaRC (**Spa**rse Plus Low-**R**ank **C**ompressors): For any matrix $X \in \mathbb{R}^{m \times n}$, let

$$\mathcal{C}_p(X) \overset{\text{def}}{=} \underset{\substack{Y = L + S \\ (m+n)\text{rank}(L) + \|S\|_0 \leq p}}{\arg\min} \|Y - X\|_F^2, \tag{4}$$

where $\| \cdot \|_0$ stands for the matrix 0-norm (number of non-zero entries) and $p$ is an integer which stands for the given parameter budget. The mapping $\mathcal{C}_p$ is indeed a compressor. Notice that $L = \text{SVD}_{\lfloor \frac{p}{m+n} \rfloor}(X), S = 0$ and $L = 0, S = \text{Top}_p(X)$ are feasible points of the minimization problem. Therefore, both

$$\|\mathcal{C}_p(X) - X\|_F^2 \leq \|\text{SVD}_{\lfloor \frac{p}{m+n} \rfloor}(X) - X\|_F^2 \leq \left(1 - \lfloor \frac{p}{(m+n)} \rfloor \frac{1}{\min\{m, n\}}\right)\|X\|_F^2, \tag{5}$$

$$\text{and} \quad \|\mathcal{C}_p(X) - X\|_F^2 \leq \|\text{Top}_p(X) - X\|_F^2 \leq \left(1 - \frac{p}{mn}\right)\|X\|_F^2, \tag{6}$$

verify $\mathcal{C}_p$ are compressors.

## 3.3 A QUALITY METRIC FOR COMPRESSORS

We analyze the single-step loss dynamics of Framework 3. The following theorem provides a bound on the expected loss after a single update step using a generic compressor $\mathcal{C}$.

**Theorem 3.1.** *Assume the loss function $\mathcal{L}$ is $L$-smooth. For an update $W_1 = W_0 - \eta g'_\xi$, the expected loss $\mathbb{E}[\mathcal{L}(W_1)]$ is bounded as follows:*

$$\mathbb{E}[\mathcal{L}(W_1)] \leq \mathcal{L}(W_0) - \frac{\eta}{2}\left(\mathbb{E}[\|g'_\xi\|^2 - (1 + \mu)\|g'_\xi - g_\xi\|^2] + \|g\|^2 - \left(1 + \frac{1}{\mu}\right)\sigma^2\right)$$

$$+ \frac{\eta^2 L}{2}\mathbb{E}[\|g'_\xi\|^2], \tag{7}$$

*where $\sigma^2 \overset{\text{def}}{=} \mathbb{E}[\|g_\xi - g\|^2]$ is the variance of the stochastic gradient and $\mu > 0$ is an arbitrary constant.*

A standard choice of $\mu$ is 1. Theorem 3.1 reveals the condition for guaranteed loss descent. By selecting a sufficiently small step size $\eta$, the final $\eta^2$ term becomes negligible. A decrease in expected loss ($\mathbb{E}[\mathcal{L}(W_1)] < \mathcal{L}(W_0)$) is then guaranteed under the bounded variance ($\sigma^2 < +\infty$) and bounded initial gradient ($\|g\|^2 < +\infty$) assumption (detailed derivations are in Appendix A.1). This gives us a sufficient condition for one-step loss decreasing if the following inequality holds:

$$\mathbb{E}[\|g'_\xi\|^2 - (1 + \mu)\|g'_\xi - g_\xi\|^2] > -\|g\|^2 + \left(1 + \frac{1}{\mu}\right)\sigma^2. \tag{8}$$

This inequality provides the crucial insight for our work. The right-hand side represents a fixed convergence barrier determined by the properties of the full gradient ($g$) and the stochastic noise ($\sigma^2$). To satisfy the condition and ensure a decrease in loss, we must choose a compressor $\mathcal{C}$ that maximizes the term on the left-hand side.

This directly motivates our metric for compressor performance. We define the Compressor Quality Metric $\mathcal{M}(\mathcal{C})$ as

$$\mathcal{M}(\mathcal{C}) \overset{\text{def}}{=} \mathbb{E}_\xi[\|\mathcal{C}(g_\xi)\|^2 - (1 + \mu)\|\mathcal{C}(g_\xi) - g_\xi\|^2]. \tag{9}$$

For a given parameter budget, the optimal compressor is the one that yields the highest value of $\mathcal{M}(\mathcal{C})$, as it provides the largest "push" against the descent barrier.

This metric has a clear interpretation related to noise robustness. High gradient variance ($\sigma^2$) is a primary cause of unstable training and divergence (Karimireddy et al., 2019). Our metric $\mathcal{M}(\mathcal{C})$ evaluates a compressor's variance-suppression ability by balancing two competing goals: preserving the gradient signal (maximizing $\|\mathcal{C}(g_\xi)\|^2$) while minimizing the compression error (minimizing $\|\mathcal{C}(g_\xi) - g_\xi\|^2$). A high-quality compressor, as measured by $\mathcal{M}(\mathcal{C})$, is therefore one that effectively retains the true gradient signal while being robust to the corrupting influence of stochastic noise.

**Algorithm 1** Adaptive Rank-Sparsity Search

**Require:** Matrix $M \in \mathbb{R}^{m \times n}$, parameter budget $p$, $n_{\text{iter}}$
1: $r_{\max} \leftarrow \lfloor \frac{p}{m+n} \rfloor$
2: $U, \Sigma, V^\top \leftarrow \text{SVD}_{r_{\max}}(M)$
3: left $\leftarrow 0$, right $\leftarrow r_{\max}$
4: **while** left $<$ right **do**
5: $\quad m_1 \leftarrow \text{left} + \lfloor (\text{right} - \text{left})/2 \rfloor$
6: $\quad m_2 \leftarrow m_1 + 1$
7: $\quad s_1 \leftarrow p - m_1(m + n)$
8: $\quad s_2 \leftarrow p - m_2(m + n)$
9: $\quad \_, \_, \text{loss}_1 \leftarrow \text{ALTPROJ}(M, m_1, s_1, 1, U, \Sigma, V)$
10: $\quad \_, \_, \text{loss}_2 \leftarrow \text{ALTPROJ}(M, m_2, s_2, 1, U, \Sigma, V)$
11: $\quad$ **if** $\text{loss}_1 < \text{loss}_2$ **then**
12: $\quad\quad$ right $\leftarrow m_1$
13: $\quad$ **else**
14: $\quad\quad$ left $\leftarrow m_2$
15: $\quad$ **end if**
16: **end while**
17: $r^* \leftarrow \text{left}$
18: $s^* \leftarrow p - r^*(m + n)$
19: $L_{\text{final}}, S_{\text{final}}, \_ \leftarrow \text{ALTPROJ}(M, r^*, s^*, n_{\text{iter}})$
20: **return** $L_{\text{final}}, S_{\text{final}}$

**Algorithm 2** Alternating Projection Method

1: **procedure** ALTPROJ($M, r, s, n_{\text{iter}}, U, \Sigma, V$)
2: $\quad S_0 \leftarrow \mathbf{0}$
3: $\quad$ **for** $k = 0, \ldots, n_{\text{iter}} - 1$ **do**
4: $\quad\quad$ **if** $k = 0$ and $U$ is provided **then**
5: $\quad\quad\quad L_{k+1} \leftarrow U_{:,:r}\Sigma_{:r,:r}V_{:,:r}^\top$
6: $\quad\quad$ **else**
7: $\quad\quad\quad L_{k+1} \leftarrow \text{SVD}_r(M - S_k)$
8: $\quad\quad$ **end if**
9: $\quad\quad S_{k+1} \leftarrow \text{Top}_s(M - L_{k+1})$
10: $\quad$ **end for**
11: $\quad L \leftarrow L_{n_{\text{iter}}}, S \leftarrow S_{n_{\text{iter}}}$
12: $\quad \text{loss} \leftarrow \|M - L - S\|_F^2$
13: $\quad$ **return** $L, S, \text{loss}$
14: **end procedure**

## 3.4 Algorithm for Solving SpaRC

To compress $g_{\xi,\ell}$ with a parameter budget $p$, we shall solves the following optimization problem by setting $M = g_{\xi,\ell}$:

$$\min_{L,S} \quad \|S + L - M\|_F^2$$
$$\text{s.t.} \quad (m+n)\text{rank}(L) + \|S\|_0 \le p, \tag{10}$$

where $S, L \in \mathbb{R}^{m \times n}$. The parameter budget $p$ is a flexible value, which could represent the total parameters available for adapters in a linear layer. For simplicity, it can be set relative to a maximum rank $r_{\max}$, e.g., $p = r_{\max}(m + n)$; or to a maximum percentage $s_{\text{ratio}}$, e.g., $p = s_{\text{ratio}}mn$.

As shown in Algorithm 1, we reformulate it as a series of subproblems by fixing the rank budget of the low-rank component, $\text{rank}(L) \le r_L$:

$$\min_{L,S} \quad \|S + L - M\|_F^2$$
$$\text{s.t.} \quad \text{rank}(L) \le r_L, \quad \|S\|_0 \le p - r_L(m + n). \tag{11}$$

We solve this subproblem using a single iteration of an alternating projection. First, the low-rank matrix $L$ is found by computing the best rank $\le r_L$ approximation of $M$. Then, the sparse matrix $S$ is found by taking the largest magnitude entries of the residual $M - L$.

Crucially, this process is highly efficient. The expensive Singular Value Decomposition (SVD) of the target matrix $M$ is performed only once as a pre-computation step. For any given rank $r_L$ in the subproblem, the optimal $L$ is constructed by simply taking a slice of the top $r_L$ singular values and vectors from this pre-computed decomposition. This reduces the SVD overhead to a fixed, one-time cost. While brute-force enumeration over all possible ranks $r_L$ is computationally prohibitive, we empirically observe that the reconstruction error from this single-step projection is unimodal with respect to the rank $r_L$. As detailed in Appendix C.2, the error curve exhibits a single, well-defined minimum. This property allows us to find the optimal rank, $r_{\text{final}}$, using an efficient binary-like search, as detailed in Algorithm 1.

## 3.5 RA-SpaRC Initialization

Given the decomposed matrices $L_{\text{final}}$ and $S_{\text{final}}$, our goal is to initialize trainable adapters that approximate this update. To properly scale this update for initialization, we introduce a scalar $\gamma$. This

allows us to interpret the process as a single-step compressed SGD, where $\frac{1}{\gamma}$ serves as the learning rate $\eta$ in Framework 3.

The sparse adapter is initialized directly. We store the non-zero values of $\frac{1}{\gamma}S_{\text{final}}$ as $E_0$.

The low-rank component requires a more nuanced approach. Inspired by LoRA (Hu et al., 2022) and SLTrain (Han et al., 2024), we seek to apply a unique scaling $\lambda$ ($= \frac{\alpha}{r}$ or $= \frac{\alpha}{\sqrt{r}}$, where $\alpha$ is the LoRA alpha) to the low-rank component to control its training speed. However, a naive scaling of the initial matrix (e.g., using $\lambda \cdot \frac{1}{\gamma}L_{\text{final}}$) is not viable, as it would violate the integrity of our initial gradient approximation.

To resolve this, we employ a reparameterization trick. While the effective update from the low-rank adapters is $\lambda BA$, we initialize the trainable matrices $B_0$ and $A_0$ as:

$$B_0 = \frac{1}{\sqrt{\lambda}}U\sqrt{\Sigma} \quad \text{and} \quad A_0 = \frac{1}{\sqrt{\lambda}}\sqrt{\Sigma}V^T, \tag{12}$$

where $U\Sigma V^T = \frac{1}{\gamma}L_{\text{final}}$ is the SVD decomposition. This design elegantly achieves two simultaneous goals. First, at initialization, the total update correctly reconstructs the target: $\lambda B_0 A_0$ equals $\frac{1}{\gamma}L_{\text{final}}$. Second, during training with a base learning rate $\eta_{\text{tr}}$, the effective learning rate for the low-rank component is precisely scaled to $\eta_{\text{tr}}\lambda$. This provides explicit control over the learning dynamics without compromising the initial state. A formal proof of this property is provided in Lemma A.3.

The final initialization result $W_1$ is

$$W_1 = W_0 - \lambda B_0 A_0 - E_0 = W_0 - \frac{1}{\gamma}(S_{\text{final}} + L_{\text{final}}) = W_0 - \eta \cdot \mathcal{C}_p(g_\xi).$$

## 4 EXPERIMENTS

In this section, we shall evaluate RA-SpaRC from various perspectives. We conduct our primary experiments on two model families, LLaMA-2-7B (Touvron et al., 2023) and the more recent Qwen2.5-7B (QwenTeam, 2024), to ensure broad applicability. Our experimental setup (as detailed in Appendix D), including data preprocessing, follows the way established in LoRA-GA (Wang et al., 2024). We evaluate our method from the following three aspects:

- Task Performance: We first assess performance of RA-SpaRC on a diverse set of Natural Language Understanding (NLU) and Generation (NLG) benchmarks.

- Compressor Comparison: Next, we conduct a direct comparison of our SpaRC compressor against standard SVD and TopK baselines to quantify its effectiveness.

- Resource Cost: Finally, we analyze the computational resource costs to demonstrate the practical efficiency of our method.

### 4.1 EXPERIMENTS ON NATURAL LANGUAGE UNDERSTANDING

Table 1 shows the T5-base fine-tuning results on a GLUE subset. All LoRA-based methods use a rank of $r = 8$, while our RA-SpaRC uses the parameter budget $r_{\text{max}} = 8$.

RA-SpaRC achieves the highest average accuracy (88.75%), outperforming all competitors on the larger datasets (MNLI, SST-2, QNLI). It trails LoRA-One marginally on the smaller CoLA and MRPC datasets, which we attribute to the higher complexity of robust adaptation model. The sparse plus low-rank structure is more challenging to optimize on limited training data. Nevertheless, the state-of-the-art average score confirms the overall effectiveness of our approach.

### 4.2 EXPERIMENTS ON NATURAL LANGUAGE GENERATION

We evaluate our method RA-SpaRC on two core capabilities: mathematical reasoning, code generation. For each task, we fine-tune both the LLaMA-2-7B and Qwen2.5-7B models and evaluate their performance on standard benchmarks.

Table 1: Accuracy comparison on GLUE subset among typical LoRA based algorithms and our RA-SpaRC. Results are reported as accuracy (%) with standard deviations over 3 runs (best in **bold**). The results marked with (∗) are sourced from Zhang et al. (2025) under the same setting.

| Method | MNLI | SST-2 | CoLA | QNLI | MRPC | Avg. |
|---|---|---|---|---|---|---|
| LoRA∗ | $85.30_{\pm 0.04}$ | $94.04_{\pm 0.09}$ | $72.84_{\pm 1.25}$ | $93.02_{\pm 0.07}$ | $68.38_{\pm 0.01}$ | 82.72 |
| LoRA+∗ | $85.81_{\pm 0.09}$ | $93.85_{\pm 0.24}$ | $77.53_{\pm 0.20}$ | $93.14_{\pm 0.03}$ | $74.43_{\pm 1.39}$ | 84.95 |
| PiSSA∗ | $85.75_{\pm 0.07}$ | $94.07_{\pm 0.06}$ | $74.27_{\pm 0.39}$ | $93.15_{\pm 0.14}$ | $76.31_{\pm 0.51}$ | 84.71 |
| LoRA-GA∗ | $85.70_{\pm 0.09}$ | $94.11_{\pm 0.18}$ | $80.57_{\pm 0.20}$ | $93.18_{\pm 0.06}$ | $85.29_{\pm 0.24}$ | 87.77 |
| LoRA-Pro∗ | $86.03_{\pm 0.19}$ | $94.19_{\pm 0.13}$ | $81.94_{\pm 0.24}$ | $93.42_{\pm 0.05}$ | $86.60_{\pm 0.14}$ | 88.44 |
| LoRA-One∗ | $85.89_{\pm 0.08}$ | $94.53_{\pm 0.13}$ | $\mathbf{82.04}_{\pm 0.22}$ | $93.37_{\pm 0.02}$ | $\mathbf{87.83}_{\pm 0.37}$ | 88.73 |
| RoSA | $85.70_{\pm 0.14}$ | $94.07_{\pm 0.29}$ | $79.71_{\pm 0.20}$ | $93.33_{\pm 0.11}$ | $77.29_{\pm 1.17}$ | 86.02 |
| RA-SpaRC | $\mathbf{86.07}_{\pm 0.12}$ | $\mathbf{94.76}_{\pm 0.38}$ | $82.01_{\pm 0.33}$ | $\mathbf{93.45}_{\pm 0.12}$ | $87.50_{\pm 0.40}$ | **88.75** |

Table 2: Comparison of our method against various fine-tuning baselines on LLaMA-2-7B and Qwen2.5-7B. We report mean accuracy (± std. dev.) on GSM8K and HumanEval. The upward arrow (↑) indicates higher is better. The best-performing method for each model is highlighted in **bold**.

| Model | Method | Params (%) | GSM8K | HumanEval |
|---|---|---|---|---|
| | LoRA | 0.297% | $59.26 \pm 0.99$ | $25.85 \pm 1.75$ |
| | LoRA-GA | 0.297% | $56.44 \pm 1.15$ | $26.95 \pm 1.30$ |
| | LoRA-One | 0.297% | $60.44 \pm 0.17$ | $28.66 \pm 0.39$ |
| | RoSA | 0.297% | $59.51 \pm 0.23$ | $25.20 \pm 0.76$ |
| LLaMA-2-7B | RoSA | 1.187% | $61.18 \pm 0.76$ | $29.26 \pm 1.21$ |
| | RoSA | 4.746% | $60.95 \pm 0.76$ | $30.79 \pm 0.91$ |
| | RA-SpaRC | 0.297% | $60.67 \pm 0.13$ | $29.88 \pm 0.87$ |
| | RA-SpaRC | 1.187% | $61.80 \pm 0.11$ | $31.50 \pm 0.57$ |
| | RA-SpaRC | 4.746% | $\mathbf{62.02 \pm 0.23}$ | $\mathbf{35.57 \pm 1.04}$ |
| | LoRA | 0.200% | $81.61 \pm 0.71$ | $68.50 \pm 1.25$ |
| | LoRA-GA | 0.200% | $81.99 \pm 0.69$ | $69.92 \pm 1.88$ |
| | LoRA-One | 0.200% | $84.43 \pm 0.13$ | $71.75 \pm 0.29$ |
| | RoSA | 0.050% | $81.20 \pm 0.50$ | $65.65 \pm 0.29$ |
| Qwen2.5-7B | RoSA | 0.100% | $81.35 \pm 0.21$ | $66.67 \pm 0.58$ |
| | RoSA | 0.200% | $81.65 \pm 0.88$ | $67.28 \pm 0.29$ |
| | RA-SpaRC | 0.050% | $84.15 \pm 0.55$ | $67.27 \pm 0.29$ |
| | RA-SpaRC | 0.100% | $84.53 \pm 0.54$ | $68.50 \pm 1.04$ |
| | RA-SpaRC | 0.200% | $\mathbf{85.06 \pm 0.21}$ | $\mathbf{72.35 \pm 0.29}$ |

- **Mathematical Reasoning:** For the math task, we fine-tune the models on a 100k sample from the MetaMathQA dataset (Yu et al., 2023). The models are then evaluated on the GSM8K test set (Cobbe et al., 2021), and we report accuracy as the primary metric.

- **Code Generation:** For the coding task, we fine-tune the models on a 100k subset of the CodeFeedback dataset (Zheng et al., 2024). We then test them on the HumanEval benchmark (Chen et al., 2021a), reporting the PASS@1 metric.

As shown in Table 2, our method consistently outperforms other leading fine-tuning techniques. This strong performance stems from our novel initialization strategy, which is specifically designed to unlock the full potential of sparse plus low-rank fine-tuning, surpassing previous initialization methods.

To demonstrate this, we first compare our method against LoRA-One, the current state-of-the-art for LoRA initialization. On LLaMA-2-7B, at an identical 0.297% parameter budget, our approach achieves a GSM8K score of 60.67 and a HumanEval score of 29.88, outperforming LoRA-One

Table 3: Summary of hyperparameter configurations for equivalent budget comparisons. For a detailed breakdown of configurations across all parameter budgets, please see Appendix D.1.

| Method | LLaMA-2-7B (0.297%) | Qwen2.5-7B (0.200%, MLP only) |
|---|---|---|
| LoRA | $r = 8$ | $r = 8$ |
| RoSA | $r = 4, s_{\text{ratio}} = 0.0015$ | $r = 4, s_{\text{ratio}} = 0.0013$ |
| RA-SpaRC | $r_{\text{max}} = 8$ | $r_{\text{max}} = 8$ |

by an absolute margin of +0.23 and +1.22 points, respectively. This advantage is confirmed on Qwen2.5-7B, where our method, using a targeted MLP-only strategy at a 0.20% budget, scores 85.06 on GSM8K and 72.35 on HumanEval, yielding improvements of +0.63 and +0.60 points over LoRA-One. Furthermore, we compare our method's scalability against RoSA, a prior method that also combines sparse plus low-rank updates. Across various parameter budget, our approach consistently delivers superior results.

Further details on the adaptive parameter budget allocations that lead to these results are provided in Appendix C.3. This analysis demonstrates the method's capability to discover effective configurations, validating the core mechanism of our approach.

### 4.3 COMPARISON OF DIFFERENT COMPRESSORS

We evaluate our compressor, SpaRC, against SVD and TopK baselines on the CodeFeedback and MetaMathQA datasets. Performance is measured by our quality metric $\mathcal{M}(\mathcal{C})$ with $\mu = 1$ and relative reconstruction error at two parameter budgets (0.297% and 1.187%).

Table 4: Comparison of different compressors. Based on LLaMA2-7B in CodeFeedback and Meta-MathQA dataset. The arrow ↑ / ↓ indicates higher/lower is better.

| Method | $\mathcal{M}(\mathcal{C})_{\text{code}}$ ↑ | $\mathbb{E}\left[\frac{\|\mathcal{C}(g_\xi) - g_\xi\|^2}{\|g_\xi\|^2}\right]_{\text{code}}$ ↓ | $\mathcal{M}(\mathcal{C})_{\text{math}}$ ↑ | $\mathbb{E}\left[\frac{\|\mathcal{C}(g_\xi) - g_\xi\|^2}{\|g_\xi\|^2}\right]_{\text{math}}$ ↓ |
|---|---|---|---|---|
| SpaRC (0.297%) | **6.99** | **18.42%** | **34.98** | **8.90%** |
| SVD (0.297%) | 6.78 | 18.89% | 34.75 | 9.07% |
| TopK (0.297%) | 0.69 | 31.68% | 10.06 | 26.34% |
| SpaRC (1.187%) | **11.69** | **9.61%** | **38.11** | **3.86%** |
| SVD (1.187%) | 11.56 | 9.88% | 38.00 | 3.94% |
| TopK (1.187%) | 3.78 | 25.47% | 16.11 | 20.81% |

The results in Table 4 reveal a clear performance hierarchy. Both SpaRC and SVD vastly outperform TopK, achieving a quality metric that is an order of magnitude higher on CodeFeedback and 3-4 times higher on MetaMathQA, along with substantially lower reconstruction error. Furthermore, SpaRC consistently maintains a slight edge over SVD in all configurations. This relative ranking (SpaRC > SVD ≫ TopK) holds across both datasets and budgets, confirming the robustness of our findings and validating SpaRC as the most effective compressor.

### 4.4 RESOURCE COSTS

Fine-tuning with sparse matrices on GPUs introduces significant computational overhead. For instance, existing methods like RoSA (Nikdan et al., 2024) exhibit a 1.7x to 2x increase in training time compared to the standard LoRA baseline, a finding we reproduce in our experiments (Figure 1). To address this bottleneck, our optimized implementation for the sparse adapter (detailed in Appendix B.1) improves efficiency. As a result, our method's training time is only 1.05x to 1.35x that of LoRA, depending on the percentage of trainable parameters. This represents a 30-40% reduction in the training overhead common to prior sparse methods.

Table 5 shows the other resource costs. Our method uses the same peak GPU memory and number of trainable parameters as LoRA. The only trade-off is a one-time initialization cost. For instance,

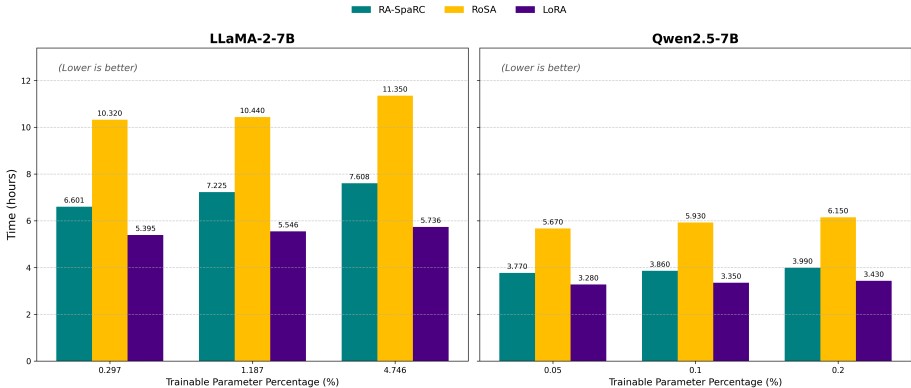

Figure 1: Training time comparison of RA-SpaRC, RoSA, and LoRA (with different initialization methods).

Table 5: Resource consumption comparison on the GSM8K dataset. Peak GPU memory was measured during training with a batch size of 1, 32 gradient accumulation steps, a sequence length of 1024, and a rank of 8.

| Model | Method | Params (%) | Peak Mem (GB) | Init Time (min) |
|---|---|---|---|---|
| LLaMA-2-7B | LoRA-One | 0.297% | 17.5 | 1.5 |
| | RoSA | 0.297% | 17.5 | 14.0 |
| | RA-SpaRC | 0.297% | 17.5 | 8.0 |
| Qwen-2.5-7B (MLP Only) | LoRA-One | 0.200% | 20.0 | 2.0 |
| | RoSA | 0.200% | 20.0 | 10.5 |
| | RA-SpaRC | 0.200% | 20.0 | 7.5 |

on LLaMA-2-7B, our 8-minute setup is significantly faster than RoSA's 14 minutes. While this is longer than LoRA-One's 1.5-minute setup, this cost occurs only once before training.

In summary, our method requires a small, affordable increase in training and initialization time compared to LoRA. We argue this cost is justified by the significant performance gains on downstream tasks. Compared to other sparse methods like RoSA, our approach is a much more practical and efficient solution that does not use extra memory.

## 5 CONCLUSION

In this work, we introduce RA-SpaRC, a novel initialization method for robust adaptation. The key advantage of RA-SpaRC is its principled and automated budget allocation strategy. By analyzing gradient information, it determines an effective split between sparse and low-rank components to ensure the most effective use of any given parameter budget.

Our extensive experimental results manifest that this hybrid initialization strategy fully realizes the potential of robust adaptation, yielding better performance compared to purely low-rank methods. We also demonstrate that our implementation is highly efficient, for both computational time and memory overhead. A promising avenue for future research is the development of more sophisticated algorithms to solve the core compression problem, which could lead to even greater performance.

ETHIC STATEMENT

We, the authors of this paper, have read and adhere to the ICLR Code of Ethics. Our work has been conducted in accordance with its general ethical principles, including contributing to societal well-being, upholding scientific excellence, avoiding harm, and being honest and transparent.

REPRODUCIBILITY STATEMENT

To ensure the reproducibility of our work, we have made comprehensive efforts to document all necessary details. The complete implementation details, including hyperparameter settings, model architectures, and dataset sources for all experiments presented in Section 4, are thoroughly described in Appendix D. Any assumptions and theoretical claims are formally stated and proven in Appendix A.

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

## A SUPPLEMENTARY PROOF

### A.1 PROOF OF THEOREM 3.1 AND ITS COROLLARY

**Theorem A.1.** *Assume the loss function $\mathcal{L}$ is L-smooth. For an update $W_1 = W_0 - \eta g'_\xi$, the expected loss $\mathbb{E}[\mathcal{L}(W_1)]$ is bounded as follows:*

$$\mathbb{E}[\mathcal{L}(W_1)] \leq \mathcal{L}(W_0) - \frac{\eta}{2} \left( \mathbb{E}[\|g'_\xi\|^2 - (1+\mu)\|g'_\xi - g_\xi\|^2] + \|g\|^2 - \left(1 + \frac{1}{\mu}\right) \sigma^2 \right)$$
$$+ \frac{\eta^2 L}{2} \mathbb{E}[\|g'_\xi\|^2], \tag{13}$$

*where $\sigma^2 \stackrel{\text{def}}{=} \mathbb{E}[\|g_\xi - g\|^2]$ is the variance of the stochastic gradient and $\mu > 0$ is an arbitrary constant.*

**Corollary A.2.** *If $\mathbb{E}[\|g'_\xi\|^2 - (1+\mu)\|g'_\xi - g_\xi\|^2] > -\|g\|^2 + \left(1 + \frac{1}{\mu}\right) \sigma^2$, $\sigma^2 < +\infty$ and $\|g\| < +\infty$, there exists an $\eta$ such that $\mathbb{E}[\mathcal{L}(W_1)] < \mathcal{L}(W_0)$.*

*Proof.* According to Lemma 2 in Li et al. (2021), we derive the following result:

$$\mathcal{L}(W_1) \leq \mathcal{L}(W_0) - \frac{\eta}{2}\|g\|^2 - (\frac{1}{2\eta} - \frac{L}{2})\|\eta g'_\xi\|^2 + \frac{\eta}{2}\|g'_\xi - g\|^2$$

$$= \mathcal{L}(W_0) - \frac{\eta}{2}\|g\|^2 - \frac{\eta}{2}\|g'_\xi\|^2 + \frac{L\eta^2}{2}\|g'_\xi\|^2 + \frac{\eta}{2}\|g'_\xi - g_\xi + g_\xi - g\|^2$$

$$\leq \mathcal{L}(W_0) - \frac{\eta}{2}\|g\|^2 - \frac{\eta}{2}\|g'_\xi\|^2 + \frac{L\eta^2}{2}\|g'_\xi\|^2 + \frac{(1+\mu)\eta}{2}\|g'_\xi - g_\xi\|^2$$

$$+ \frac{(1 + \frac{1}{\mu})\eta}{2}\|g_\xi - g\|^2, \tag{14}$$

$$\mathbb{E}[\mathcal{L}(W_1)] \leq \mathcal{L}(W_0) - \frac{\eta}{2}\|g\|^2 - \frac{\eta}{2}\mathbb{E}\|g'_\xi\|^2 + \frac{L\eta^2}{2}\mathbb{E}\|g'_\xi\|^2 + \frac{(1+\mu)\eta}{2}\mathbb{E}\|g'_\xi - g_\xi\|^2$$

$$+ \frac{(1 + \frac{1}{\mu})\eta}{2}\sigma^2$$

$$\leq \mathcal{L}(W_0) - \frac{\eta}{2}\mathbb{E}[\|g'_\xi\|^2 - (1+\mu)\|g'_\xi - g_\xi\|^2 + \|g\|^2 - (1 + \frac{1}{\mu})\sigma^2]$$

$$+ \frac{L\eta^2}{2}\mathbb{E}\|g'_\xi\|^2. \tag{15}$$

Let $G = 2\sigma^2 + 2\|g\|^2 < +\infty$, $\mathbb{E}[\|g_\xi\|^2] = \mathbb{E}[\|g_\xi - g + g\|^2] \leq 2\sigma^2 + 2\|g\|^2 = G$,

$$\mathbb{E}[\|g'_\xi\|^2] = \mathbb{E}[\|g'_\xi - g_\xi + g_\xi\|^2] \leq \mathbb{E}[2\|g'_\xi - g_\xi\|^2 + 2\|g_\xi\|^2] = 2(2-\alpha)\mathbb{E}[\|g_\xi\|^2] \leq 2(2-\alpha)G.$$

Denote $D = \mathbb{E}[\|g'_\xi\|^2 - (1+\mu)\|g'_\xi - g_\xi\|^2 + \|g\|^2 - (1 + \frac{1}{\mu})\sigma^2]$, based on inequality 15, we can find $\eta \in (0, \frac{D}{2LG(2-\alpha)})$ such that $\mathbb{E}[\mathcal{L}(W_1)] < \mathcal{L}(W_0)$. □

### A.2 PROOF OF DIFFERENTIATED LEARNING RATE

**Lemma A.3** (Effective Learning Rate Scaling). *Let the low-rank adapter matrices $B$ and $A$ be initialized as $B_0 = \frac{1}{\sqrt{\lambda}}B'_0$ and $A_0 = \frac{1}{\sqrt{\lambda}}A'_0$, where $B'_0 A'_0 = \frac{1}{\gamma}L_{final}$. When training with an optimizer using a learning rate $\eta$, the effective learning rate applied to the conceptual matrices $B'$ and $A'$ is exactly $\eta\lambda$.*

*Proof.* The effective update to the model weights from the low-rank adapter is given by the product $\Delta W_{\text{LoRA}} = \lambda B A$. At initialization, the parameters are $B_0$ and $A_0$.

First, we establish the relationship between the gradients. Let the loss be $\mathcal{L}$. The gradient of the loss with respect to the trainable parameter $B$ is computed via the chain rule.

$$\nabla_B \mathcal{L} = \frac{\partial \mathcal{L}}{\partial(\lambda BA)} \frac{\partial(\lambda BA)}{\partial B} = \nabla_{\Delta W} \mathcal{L} \cdot (\lambda A^T).$$

Now, consider the gradient with respect to the conceptual matrix $B'$.

$$\nabla_{B'} \mathcal{L} = \frac{\partial \mathcal{L}}{\partial(B'A')} \frac{\partial(B'A')}{\partial B'} = \nabla_{B'A'} \mathcal{L} \cdot (A')^T.$$

Since $\lambda B_0 A_0 = B_0' A_0'$, the gradient of the loss with respect to the output product is the same ($\nabla_{\Delta W_0} \mathcal{L} = \nabla_{B_0' A_0'} \mathcal{L}$). We can therefore relate the gradients of the parameters at initialization:

$$\nabla_{B_0} \mathcal{L} = \lambda(\nabla_{\Delta W_0} \mathcal{L})A_0^T = \lambda(\nabla_{\Delta W_0} \mathcal{L})\left(\frac{1}{\sqrt{\lambda}}A_0'\right)^T = \sqrt{\lambda}\left((\nabla_{\Delta W_0} \mathcal{L})(A_0')^T\right) = \sqrt{\lambda}\nabla_{B_0'} \mathcal{L}.$$

Similarly, it can be shown that $\nabla_{A_0} \mathcal{L} = \sqrt{\lambda}\nabla_{A_0'} \mathcal{L}$.

During an optimizer step, the trainable parameters $B$ and $A$ are updated as:

$$B_1 = B_0 - \eta\nabla_{B_0} \mathcal{L}, \quad A_1 = A_0 - \eta\nabla_{A_0} \mathcal{L} \tag{16}$$

To understand the effect of this update on the conceptual matrices, we define the updated conceptual matrices, $B_1'$ and $A_1'$, in terms of the updated trainable parameters, maintaining the relationship $B_1' = \sqrt{\lambda}B_1$. By substituting the update rule for $B_1$ and the gradient relationship, we get:

$$\begin{aligned} B_1' = \sqrt{\lambda}B_1 &= \sqrt{\lambda}(B_0 - \eta\nabla_{B_0} \mathcal{L}) = \sqrt{\lambda}B_0 - \eta\sqrt{\lambda}\nabla_{B_0} \mathcal{L} \\ &= B_0' - \eta\sqrt{\lambda}(\sqrt{\lambda}\nabla_{B_0'} \mathcal{L}) = B_0' - \eta\lambda\nabla_{B_0'} \mathcal{L}. \end{aligned} \tag{17}$$

The same derivation holds for $A_1'$:

$$A_1' = \sqrt{\lambda}A_1 = \sqrt{\lambda}(A_0 - \eta\nabla_{A_0} \mathcal{L}) = A_0' - \eta\lambda\nabla_{A_0'} \mathcal{L}.$$

These equations show that the update rule for the conceptual matrices $B'$ and $A'$ is precisely that of a gradient descent step with a learning rate of $\eta\lambda$. This proves that our reparameterization scales the effective learning rate for the low-rank component by the factor $\lambda$ exactly, without any approximation. This completes the proof. $\square$

# B  SYSTEM IMPLEMENTATION

## B.1  SYSTEM IMPLEMENTATION

Our implementation must efficiently compute the output for the composed weight matrix $W_0 + \text{Mat}(E) + \lambda BA$ and its gradients. The primary challenge is avoiding the materialization of dense matrices, particularly the full gradient tensor for the sparse component.

**Forward Pass**  Like RoSA (Nikdan et al., 2024), we handle the sparse component by adding it to the pre-trained weights $W_0$. However, our implementation uses a different data structure. While RoSA uses the Compressed Sparse Row (CSR) format, we found this less efficient for the scattered, non-row-concentrated sparsity patterns learned by our method. We therefore represent $E$ with its non-zero values ($E_{val}$) and their indices ($E_{idx}$) and apply them to a copy of $W_0$ using an optimized `torch.scatter_add_` operation. This approach is faster for our specific use case. The final output is then computed by summing the low-rank path $(xA^T)B^T$ and the output from the updated weights.

**Backward Pass**  The main efficiency gain comes from our custom backward kernel for the sparse component. A standard autograd approach would first materialize the entire dense gradient matrix $\nabla_{\text{Mat}(E)}\mathcal{L} = (\nabla_y \mathcal{L})^T x$, and then gather the values corresponding to the non-zero indices, $(\nabla_{\text{Mat}(E)}\mathcal{L})_{E_{idx}}$. This intermediate dense tensor is prohibitively memory-intensive.

To circumvent this, we implement a custom kernel that computes the gradient vector $\nabla_{E_{val}}\mathcal{L}$ directly, bypassing the dense matrix. For each non-zero element $E_{val}[i]$ located at matrix coordinates $(r, c)$, our kernel computes its gradient as the inner product of the corresponding columns of the upstream gradient and the input, which can be executed in parallel for all non-zero elements:

$$\nabla_{E_{val}[i]}\mathcal{L} = \langle (\nabla_y \mathcal{L})_{:,r}, x_{:,c} \rangle.$$

By fusing the gradient calculation and indexing into a single block-parallelizable kernel, we eliminate the primary memory and computational bottleneck of the backward pass, achieving significant speedups over naive implementations.

# C   SUPPLEMENTARY EXPERIMENTS

## C.1   INSTRUCTION FOLLOWING RESULTS

To evaluate performance on general knowledge and problem-solving, we fine-tune the models on the Alpaca dataset (Taori et al., 2023). We then measure the zero-shot accuracy on the Massive Multitask Language Understanding (MMLU) benchmark (Hendrycks et al., 2021). While a five-shot setting is commonly used for MMLU, we specifically use a zero-shot approach. This is because our goal is to test the model's core instruction-following ability gained from the Alpaca fine-tuning itself. A five-shot evaluation tests how well a model can learn from examples given in the prompt (in-context learning), which would make it difficult to isolate the direct impact of our fine-tuning method. The zero-shot setting provides a clearer measure of the model's generalized capabilities.

The results in Table 6 show different outcomes for the two models. For LLaMA-2-7B, all fine-tuning methods provide a clear improvement over the base model. Our method, RA-SpaRC, achieves the highest accuracy at 46.14%, showing it is very effective at improving the model's general problem-solving skills.

For Qwen2.5-7B, however, the improvements are very small. A likely reason is that the base Qwen2.5-7B model is already excellent at following instructions. It is also possible that its original training data already contained the Alpaca dataset or something very similar. If so, fine-tuning on Alpaca offers little new information, which would explain the small gains. Even with these small improvements, RA-SpaRC still achieves the highest score, showing it provides a consistent, if minor, benefit.

Table 6: Comparison of fine-tuning methods on LLaMA-2-7B and Qwen2.5-7B. Models are fine-tuned on Alpaca and evaluated with zero-shot accuracy on MMLU. We report the mean accuracy ($\pm$ std. dev.). The upward arrow ($\uparrow$) indicates higher is better. The best method for each model is in **bold**.

| Model | Method | Params (%) | MMLU Accuracy (%) $\uparrow$ |
|---|---|---|---|
| | Base Model | None | 40.79 |
| | LoRA | 0.297% | $42.84 \pm 0.12$ |
| LLaMA-2-7B | LoRA-One | 0.297% | $45.52 \pm 0.31$ |
| | RoSA | 0.297% | $44.03 \pm 0.28$ |
| | **RA-SpaRC** | **0.297%** | **$46.14 \pm 0.14$** |
| | Base Model | None | 70.50 |
| | LoRA | 0.200% | $70.53 \pm 0.04$ |
| Qwen2.5-7B | LoRA-One | 0.20% | $70.59 \pm 0.09$ |
| | RoSA | 0.200% | $70.53 \pm 0.22$ |
| | **RA-SpaRC** | **0.200%** | **$70.62 \pm 0.10$** |

## C.2   UNIMODALITY ASSUMPTION

This section provides the empirical evidence for the general unimodal behavior that underpins Algorithm 1. We demonstrate that for a fixed parameter budget, the one-step alternative projection loss for stochastic gradients exhibits a single, well-defined minimum.

**Experimental Setup.**   Our validation procedure was executed with the following precise settings:

- **Model:** We used the LLaMA-2-7B model.

- **Datasets:** Stochastic gradients are estimated on three distinct fine-tuning datasets: Meta-MathQA, CodeFeedback, and Alpaca.

- **Gradient Estimation:** For each dataset, a single stochastic gradient is computed using a mini-batch of 8 samples. This gradient matrix is the target for our decomposition.

- **Decomposition Parameters:** The decomposition is constrained by a fixed parameter budget equivalent to a dense low-rank approximation with a maximum rank of $r_{\max} = 8$. We

enumerated all integer ranks $r \in [0, r_{\max}]$. The corresponding number of sparse elements, $s$, was calculated to maintain the budget, following the relation $s = (r_{\max} - r)(m + n)$, where $m$ and $n$ are the dimensions of the gradient matrix.

- **Loss Metric:** For each $(r, s)$ pair, we computed the single-step alternating projection error.

**Results and Analysis.** To demonstrate the robustness of this property across the model's depth, we analyzed the gradients from multiple layers. For simplicity and generality, Figure 2 & 3 show the results for three representative layers: an early layer (0), a middle layer (15), and a late layer (31).

Crucially, each visualized loss landscape represents the one-step projection loss from all linear modules within that specific layer. This includes the gradients from the four attention projections (query, key, value, output) and the three MLP projections (gate, up, down). This aggregation confirms that the unimodal property is not specific to a single module but is a general characteristic of the layer's entire gradient structure.

## C.3 BUDGET ALLOCATION RESULTS

We visualize the allocation results of RA-SpaRC over different models and datasets in Figure 4. Only the rank distribution of the low-rank component for each layer is shown, as the number of non-zero elements of the sparse component can be computed by subtracting the corresponding parameters of low-rank component from the total parameter budget. One typical feature is that when the parameter budget is stringent, the solution of RA-SpaRC coincides with direct SVD in many situations. But when the parameter budget is relaxed, more patterns of the combinations of low-rank and sparse adapters are found.

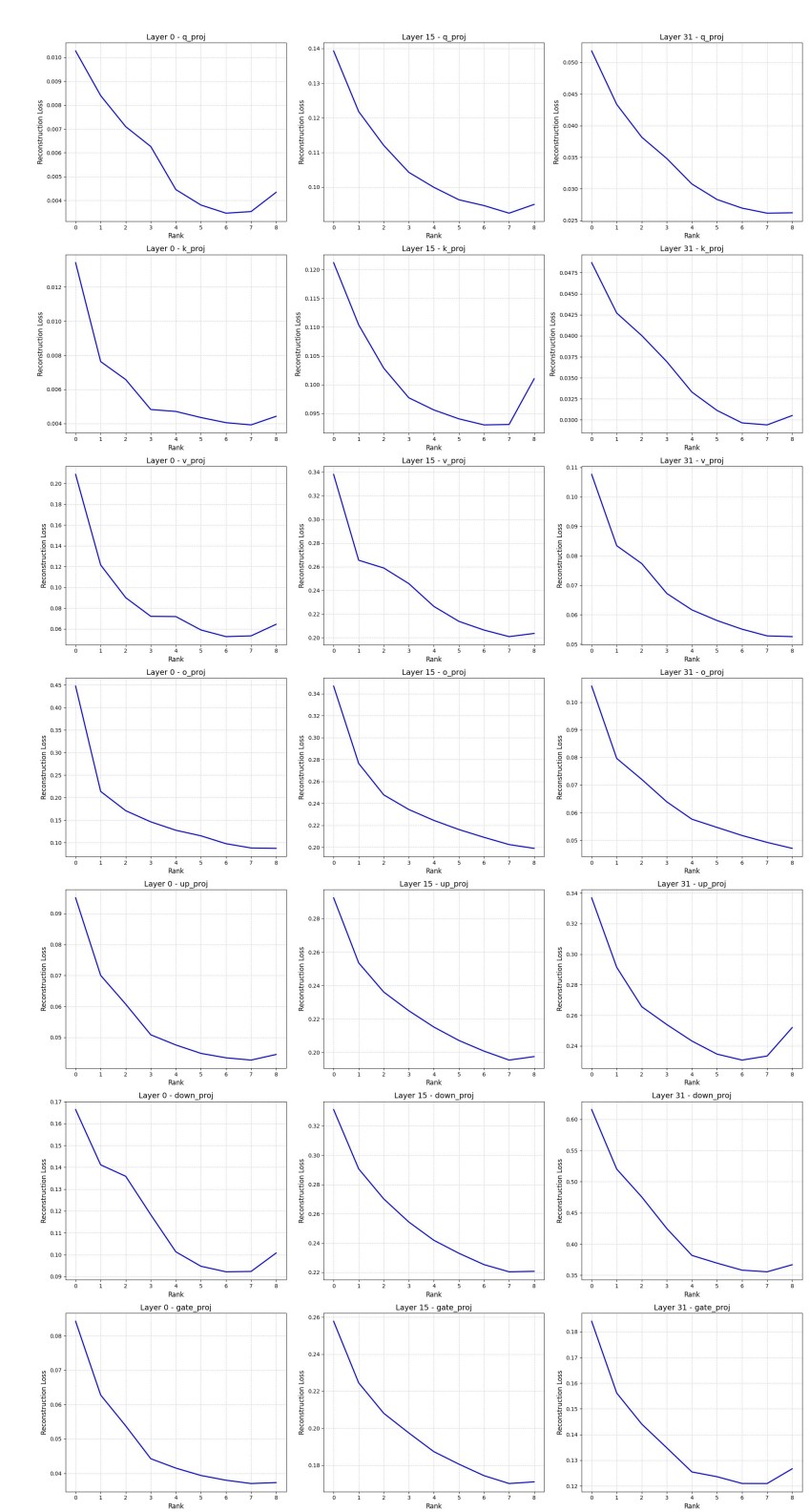

Figure 2: Loss curve for gradient decomposition on the MetaMathQA dataset.

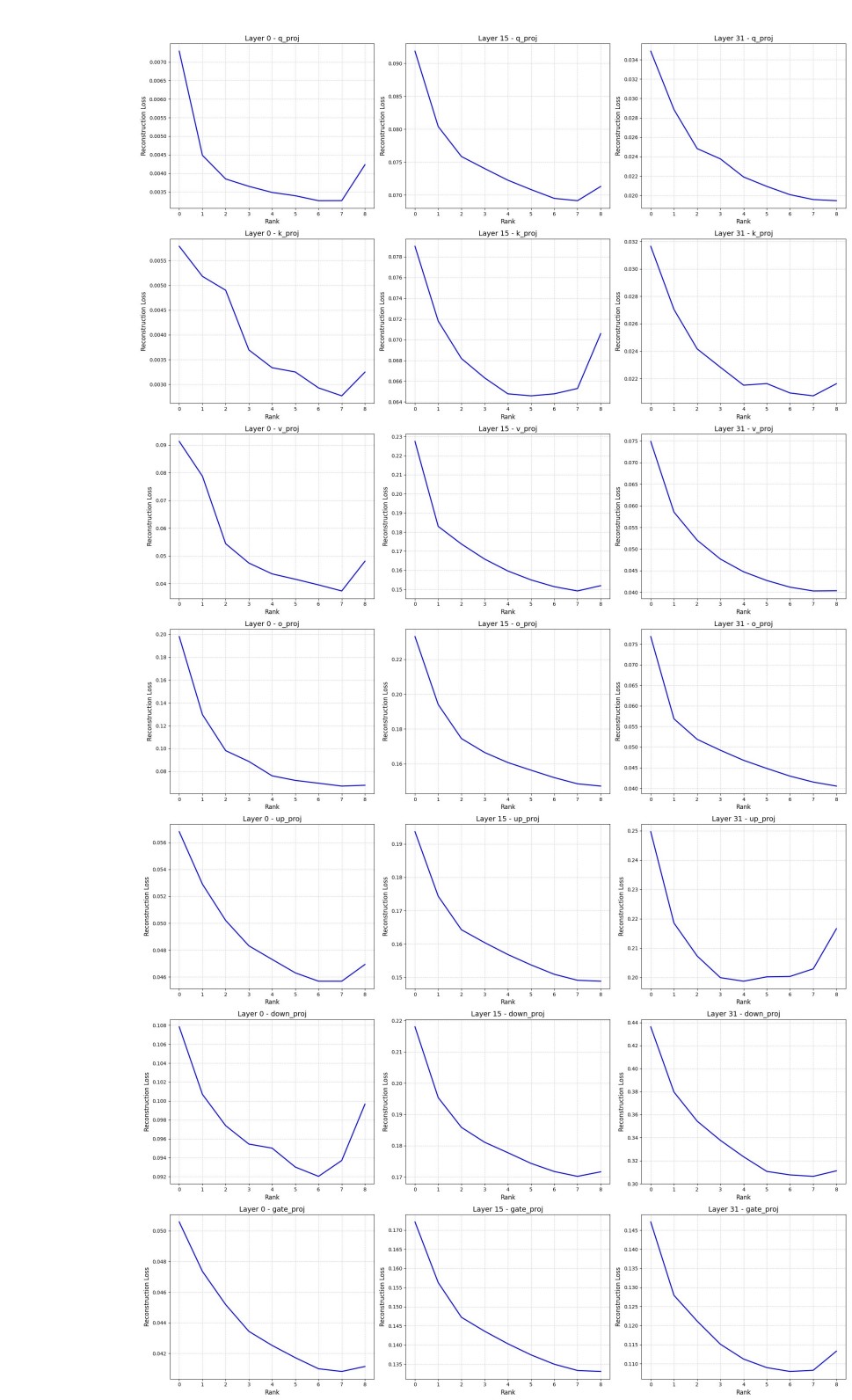

Figure 3: Loss curve for gradient decomposition on the CodeFeedback dataset.

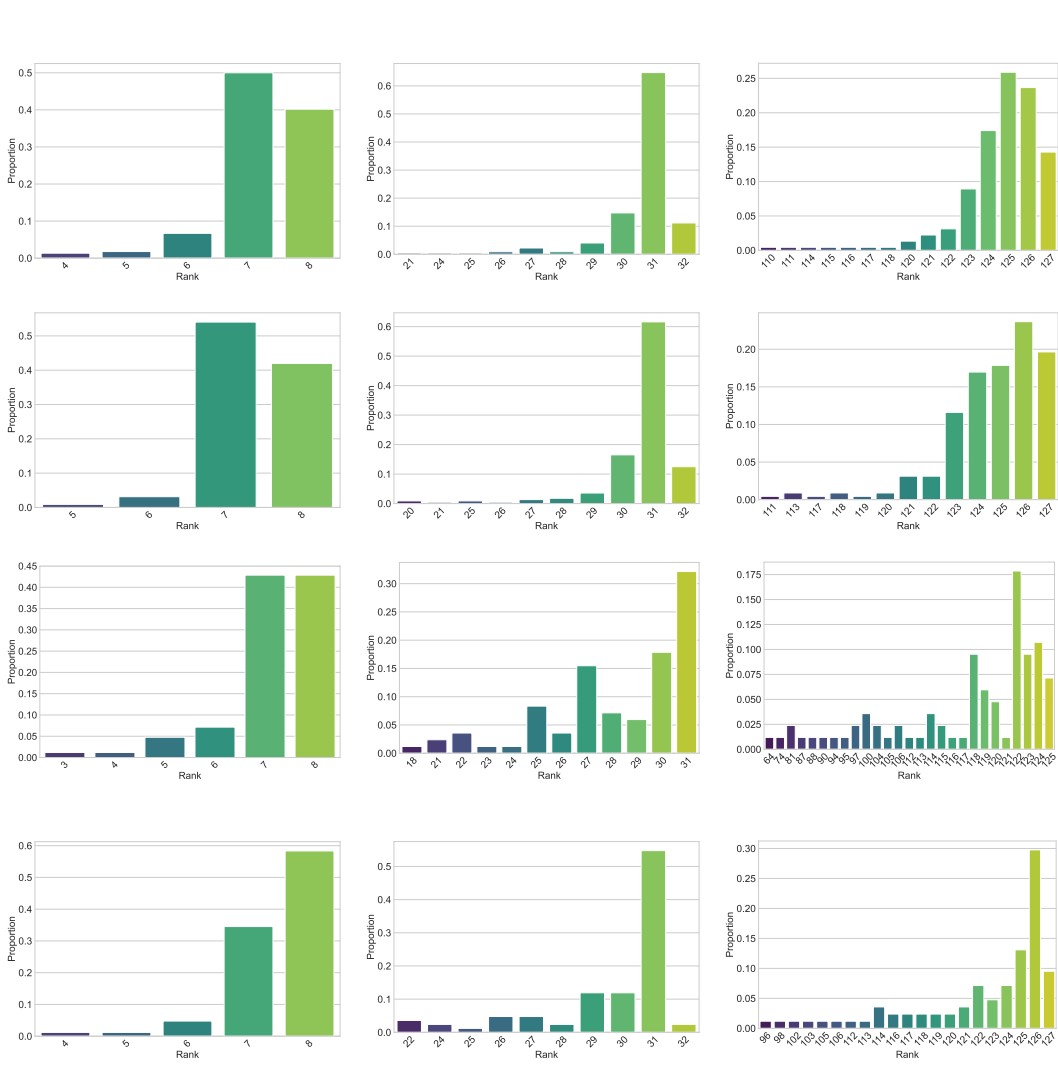

Figure 4: Rank distribution across different models and datasets using RA-SpaRC. From top to bottom: LLaMA-2-7B on CodeFeedback, LLaMA-2-7B on MetaMath, Qwen2.5-7B on CodeFeedback, and Qwen2.5-7B on MetaMath. Each row shows results with maximum ranks 8, 32, and 128 (left to right).

# D EXPERIMENTAL SETTINGS

## D.1 PARAMETER BUDGET CONFIGURATIONS

This section details the hyperparameter configurations used to achieve equivalent trainable parameter counts for the different fine-tuning methods on LLaMA-2-7B and Qwen2.5-7B.

Table 7: Hyperparameter configurations for different methods on **LLaMA-2-7B**. The configurations are set to match the parameter counts benchmarked against LoRA with ranks $r = 8, 32, 128$.

| Method | 0.297% | 1.187% | 4.746% |
|---|---|---|---|
| LoRA & Variants | $r = 8$ | $r = 32$ | $r = 128$ |
| RoSA | $r = 4$ $s_{\text{ratio}} = 0.0015$ | $r = 16$ $s_{\text{ratio}} = 0.006$ | $r = 64$ $s_{\text{ratio}} = 0.024$ |
| RA-SpaRC | $r_{\text{max}} = 8$ | $r_{\text{max}} = 32$ | $r_{\text{max}} = 128$ |

Table 8: Hyperparameter configurations for different methods on **Qwen2.5-7B**. The configurations are set to match the parameter counts benchmarked against LoRA with ranks $r = 2, 4, 8$.

| Method | 0.050% | 0.100% | 0.200% |
|---|---|---|---|
| LoRA & Variants | $r = 2$ | $r = 4$ | $r = 8$ |
| RoSA | $r = 1$ $s_{\text{ratio}} = 0.0013$ | $r = 2$ $s_{\text{ratio}} = 0.0026$ | $r = 4$ $s_{\text{ratio}} = 0.0052$ |
| RA-SpaRC | $r_{\text{max}} = 2$ | $r_{\text{max}} = 4$ | $r_{\text{max}} = 8$ |

## D.2 HYPERPARAMETER CONFIGURATIONS

This section details the hyperparameter configurations for our experiments. To ensure fair comparisons, we adapt our hyperparameter search strategy from Zhang et al. (2025).

**Implementation Details.** All fine-tuning experiments run on a single NVIDIA A100 40G SXM4 GPU. We load the T5-base model in its original FP32 precision, while the LLaMA-2-7B and Qwen2.5-7B models are loaded in BF16 precision.

**NLU Tasks (T5-base).** For the Natural Language Understanding (NLU) tasks, we fine-tune the T5-base model using our RA-SpaRC method. The common hyperparameters for this setup are in Table 9. We optimize the learning rate by performing a grid search over the set $\{1 \times 10^{-3}, 5 \times 10^{-4}, 2 \times 10^{-4}, 1 \times 10^{-4}\}$. The final, task-specific learning rates and RA-SpaRC scaling parameters ($\gamma$) are presented in Table 10.

**NLG Tasks (LLaMA-2 & Qwen2.5).** For the Natural Language Generation (NLG) tasks, we fine-tune LLaMA-2-7B and Qwen2.5-7B. The common hyperparameters for these models are in Table 11. For these tasks, we conduct a more extensive search. We search the learning rate over $\{2 \times 10^{-4}, 1 \times 10^{-4}, 5 \times 10^{-5}, 2 \times 10^{-5}\}$ and the per-device batch size over $\{16, 32, 128\}$. The final, optimal hyperparameters for each model and dataset are presented in Table 12.

Table 9: Common hyperparameters for RA-SpaRC fine-tuning on the T5-base model for NLU tasks.

| Epoch | Optimizer | $(\beta_1, \beta_2)$ | $\epsilon$ | Precision | Weight Decay |
|---|---|---|---|---|---|
| 1 | AdamW | (0.9, 0.999) | $1 \times 10^{-8}$ | FP32 | 0 |
| Warm-up Ratio | LoRA $\alpha$ | LR Scheduler | Max Length | #Runs | Gradient Batch Size |
| 0.03 | 16 | cosine | 128 | 3 | 8 |

Table 10: Final selected hyperparameters for NLU tasks on T5-base with RA-SpaRC.

| Dataset | Learning Rate | Batch Size | Scaling $\gamma$ |
|---|---|---|---|
| MNLI | $5 \times 10^{-4}$ | 32 | 128 |
| SST-2 | $5 \times 10^{-4}$ | 32 | 32 |
| CoLA | $5 \times 10^{-4}$ | 32 | 16 |
| QNLI | $5 \times 10^{-4}$ | 32 | 16 |
| MRPC | $1 \times 10^{-3}$ | 32 | 128 |

Table 11: Common hyperparameters for fine-tuning LLaMA-2-7B and Qwen2.5-7B on NLG tasks.

| Epoch | Optimizer | $(\beta_1, \beta_2)$ | $\epsilon$ | Precision | Weight Decay |
|---|---|---|---|---|---|
| 1 | AdamW | (0.9, 0.999) | $1 \times 10^{-8}$ | FP32 | 0 |
| Warm-up Ratio | LoRA $\alpha$ | LR Scheduler | Max Length | #Runs | Gradient Batch Size |
| 0.03 | 16 | cosine | 1024 | 3 | 8 |

Table 12: Final selected hyperparameters for NLG tasks with RA-SpaRC.

| Model | Dataset | Learning Rate | Batch Size | Scaling $\gamma$ |
|---|---|---|---|---|
| LLaMA-2-7B | MetaMathQA | $2 \times 10^{-4}$ | 32 | 16 |
| | CodeFeedback | $5 \times 10^{-4}$ | 32 | 16 |
| | Alpaca | $2 \times 10^{-4}$ | 32 | 16 |
| Qwen2.5-7B | MetaMathQA | $2 \times 10^{-4}$ | 32 | 16 |
| | CodeFeedback | $2 \times 10^{-4}$ | 32 | 32 |
| | Alpaca | $2 \times 10^{-4}$ | 32 | 32 |

# E   LLM USAGE STATEMENT

In the preparation of this paper, Large Language Models (LLMs) serve as a writing assistance tool. Their primary function is for proofreading and language refinement, which includes correcting grammatical errors, improving sentence structure, and enhancing the overall clarity and readability of the text.

The authors employ these models specifically for polishing the writing in the Introduction, Related Work, and Experiments sections.

Crucially, LLMs do not contribute to any aspect of research ideation, formulation of hypotheses, experimental design, data analysis, or the generation of core scientific conclusions. The conceptual framework and all intellectual contributions of this work are developed exclusively by the human authors. The authors have reviewed, edited, and take full responsibility for all content presented in this paper.

