# OpenReview forum: "RA-SpaRC: Robust Adaptation with Sparse Plus Low-Rank Compressors"
_ICLR.cc/2026/Conference — ICLR 2026 Conference Withdrawn Submission_

### Official Review · Reviewer_Yfsr · 2025-10-26

**Soundness:** 2
**Presentation:** 2
**Contribution:** 2
**Rating:** 4
**Confidence:** 4

**Summary:**

The paper introduces RA-SpaRC, a new initialization strategy for Parameter-Efficient Fine-Tuning (PEFT) that addresses the limitation of existing hybrid methods like RoSA requiring manual tuning of sparse and low-rank components.

RA-SpaRC uses an adaptive allocation mechanism based on gradient analysis to automatically balance these components within a fixed parameter budget, outperforming LoRA, its variants, and RoSA across multiple benchmarks.

**Strengths:**

The paper addresses an important problem of developing an initialization strategy that enables flexible, automatic, and effective budget allocation for Robust Adaptation.

To this end, they propose RA-SpaRC (Robust Adaptation with Sparse plus Low-Rank Compressors), an initialization strategy for sparse plus low-rank fine-tuning, which could dynamically assign the ratio of low-rank and sparse parts according to the gradient information of different tasks and models.

The proposed method is very pragmatic and practical.

The proposed method was examined in comparison with different LoRA variations in several benchmarks.

**Weaknesses:**

Notation should be revised and redundancy in some terms should be fixed.

The proposed method performs on par in some experiments compared to the state of the art LoRA variations.

The initialization and running time of the proposed method is pretty large compared to the baseline vanilla LoRA.

Theoretical analyses of several properties of the proposed method, such as the convergence rate, were not provided.

**Questions:**

Have you compared convergence rate of your proposed method and the other LoRA variations theoretically?

The accuracy boost is less for GSM8K compared to the HumanEval in Table 2. Could you please elaborate this result?

---

### Official Review · Reviewer_xqnn · 2025-10-31

**Soundness:** 2
**Presentation:** 3
**Contribution:** 3
**Rating:** 4
**Confidence:** 3

**Summary:**

The paper introduces RA-SpaRC (Robust Adaptation with Sparse plus Low-Rank Compressors), a new hybrid PEFT initialization strategy that overcomes the need to manually tune the ratio of parameters between low-rank and sparse components, through an automatic parameter allocation mechanism. The paper shows that RA-SpaRC outperforms classic and hybrid PEFT methods in extensive experiments across multiple models.

The paper is a pleasure to read and the idea seems interesting and promising, offering new insights into the performance of different PEFT initialisation methods and PEFT methods themselves. Still, the empirical improvements seem limited, and it is not fully clear if the empirical evaluation is completely fair.

**Strengths:**

-	The paper is well presented and conveys the message well.
-	The proposed method seems novel in its approach.
-	The claims are supported by theory and experiments.
-	The proposed method advances the PEFT research, introducing a new hybrid approach.

**Weaknesses:**

-	The average improvement in Table 1 is marginal, especially when compared to LoRA-One.
-	Results in Table 2 are more significant, however comparisons may not be fair when considering classic PEFT methods (i.e. LoRA, LoRA-GA, LoRA-One), as for LLaMA-2 results are not reported for 4.746% parameters, but only for 0.297%, which marginally improves for GSM8K. In addition, Table 2 omits results for PISSA, which seems a relevant competitor.
-	In Section 4.4 the paper acknowledges the initialization and training times as core limitations of the proposed method, where the overhead is respectively 3.75x to 5.33x (depending on the model) and 1.05x to 1.35x. The paper fails to show the trade-offs between performance gains and time overhead, which should be the overarching goal.

**Questions:**

- how does PISSA compare?
- is there a favourable time-accuracy trade-off for the proposed method?
- why are the models (Sect. 4.2) fine-tuned only on a sample?
- what is the accuracy for the experiments in Fig. 2/3?

---

### Official Review · Reviewer_YjV5 · 2025-10-31

**Soundness:** 3
**Presentation:** 3
**Contribution:** 2
**Rating:** 4
**Confidence:** 3

**Summary:**

The paper proposes RA-SpaRC, a novel initialization strategy for parameter-efficient fine-tuning that automatically balances sparse and low-rank components when adapting large pretrained models. Concretely, RA-SpaRC adopts a compressor-based framework and defines a compressor which optimimally decomposes gradient updates into sparse and low-rank components under a fixed computational budget. It further contributes an efficient alternating projection algorithm to automatically determine the best rank–sparsity trade-off as well as a compressor quality metric that guarantees loss reduction during optimization.

Experiments on LLaMA-2-7B, Qwen2.5-7B, and T5-base models across NLU, NLG, and code reasoning tasks show consistent improvements over LoRA, LoRA-One, and RoSA, without extra memory and with moderate initialization cost.

**Strengths:**

* The proposed method identifies the shortcomings of fixed-ratio sparse + low-rank hybrid PEFT methods and comes up with an elegant formulation to adaptively tune this.
* The paper reformulates PEFT initialization as applying a compressor on gradients, unifying sparse and low-rank initialization under a unified lens.
* The authors implement a custom kernel for avoiding the materialization of dense matrices, therefore achieving real speedup compared to naive unoptimized implementations.

**Weaknesses:**

* The evaluation has the potential to be improved.
    - First off, I would recommend that a more complete experimental setup is provided, which would enhance understanding and reproducibility.
    - Breadth-wise, results are mainly focused on dense LLMs and text-based tasks. Applying the technique to other networks, e.g., ViTs, would help showcase the generality of the method.
    - Depth-wise, the paper’s evaluation section mostly focuses on aggregate results (accuracy, efficiency) rather than offering deeper insights into why or how the adaptive rank–sparsity allocation interacts with different layers of large language models (LLMs). Coupling this with different budgets would be even better.
* The paper focuses on better initialization, but adaptation dynamics during training (e.g., stability or convergence rate) are not deeply analyzed.
* Though the initialization is efficient, the adaptive search may still be costly for very large models or frequent reinitialization (e.g. adapter soups).

**Questions:**

* How would the authors propose their solution be applied on non-dense, multi-branch models, like MoE or hybrid-attention structures?
* How does the method perform under quantized pretrained backbones?
* How does RA-SpaRC interact with alternative modern optimizers (e.g., Muon)? Is the initialization's benefit amplified or diminished?
* How does the technique behave against DoRA?
* Does adaptive allocation generalize across tasks, or must it be recomputed for every fine-tuning run?
* Could the compressor be integrated dynamically during training rather than only at initialization?

---

### Official Review · Reviewer_39B7 · 2025-11-01

**Soundness:** 2
**Presentation:** 2
**Contribution:** 2
**Rating:** 2
**Confidence:** 3

**Summary:**

This paper proposes RA-SpaRC, which first computes the initial gradient on a small data batch and, under a fixed parameter budget, employs a binary search to explore different ranks. It allocates the remaining budget to sparsity, selects the rank $ r^* $ that minimizes reconstruction error, and constructs the sparse matrix ( S ) from the $ s^* $ largest-magnitude elements of the residual.

**Strengths:**

1. The proposed method can automatically allocate the ratio between low rank and sparsity which reduce the complexity of hyper-parameter fine-tuning.

2. The paper shows many results on various benchmark with different models, which demonstrate it is better than LoRA and its variants.

**Weaknesses:**

1. Since the paper determines the ratio between low-rank and sparse components based on a mini-batch, it should demonstrate whether this ratio remains stable or changes when the samples in the mini-batch vary.

2. From Tables 1 and 2, the performance gap between RA-SpaRC and LoRA-One is minor, while RA-SpaRC incurs higher initialization time. This raises questions about its practical benefit. Moreover, Table 2 should include results for LoRA-One using the same number of parameters as RA-SpaRC when evaluated on LLaMA-2-7B.

3. In Table 4, the performance difference between SpaRC and SVD is minimal. The authors should therefore provide experimental results for the proposed method without the sparse compressor to better isolate its contribution.

4. Since RoSA is the main baseline for comparison, the paper should report results for RoSA under additional configurations to more clearly demonstrate the effectiveness of the proposed adaptive ratio allocation.

**Questions:**

See the weakness

---

### Note · Authors · 2025-11-22

I have read and agree with the venue's withdrawal policy on behalf of myself and my co-authors.